# Budbreak patterns and phytohormone dynamics reveal different modes of action between hydrogen cyanamide- and defoliant-induced flower budbreak in blueberry under inadequate chilling conditions

**Syuan-You Lin**, **Shinsuke Agehara***

Gulf Coast Research and Education Center, Institute of Food and Agricultural Sciences, University of Florida, Gainesville, Florida, United States of America

\* sagehara@ufl.edu

**Data Availability Statement:** All relevant data are within the manuscript and its Supporting Information files.

## Abstract

Under inadequate chilling conditions, hydrogen cyanamide (HC) is often used to promote budbreak and improve earliness of Southern highbush blueberry (*Vaccinium corymbosum* L. interspecific hybrids). However, HC is strictly regulated or even banned in some countries because of its high hazardous properties. Development of safer and effective alternatives to HC is critical to sustainable subtropical blueberry production. In this study, we examined the efficacy of HC and defoliants as bud dormancy-breaking agents for 'Emerald' blueberry. First, we compared water control, 1.0% HC (9.35 L ha$^{-1}$), and three defoliants [potassium thiosulfate (KTS), urea, and zinc sulfate (ZS)] applied at 6.0% (28 kg ha$^{-1}$). Model fitting analysis revealed that only HC and ZS advanced both defoliation and budbreak compared with the water control. HC-induced budbreak showed an exponential plateau function with a rapid phase occurring from 0 to 22 days after treatment (DAT), whereas ZS-induced budbreak showed a sigmoidal function with a rapid phase occurring from 15 to 44 DAT. The final budbreak percentage was similar in all treatments (71.7%–83.7%). Compared with the water control, HC and ZS increased yield by up to 171% and 41%, respectively, but the yield increase was statistically significant only for HC. Phytohormone profiling was performed for water-, HC- and ZS-treated flower buds. Both chemicals did not increase gibberellin 4 and indole-3-acetic acid production, but they caused a steady increase in jasmonic acid (JA) during budbreak. Compared with ZS, HC increased JA production to a greater extent and was the only chemical that reduced abscisic acid (ABA) concentrations during budbreak. A follow-up experiment tested ZS at six different rates (0–187 kg ha$^{-1}$) but detected no significant dose-response on budbreak. These results collectively suggest that defoliants are not effective alternatives to HC, and that HC and ZS have different modes of action in budbreak induction. The high efficacy of HC as a dormancy-breaking agent could be due to its ability to reduce ABA concentrations in buds. Our results also suggest that JA accumulation is involved in budbreak induction in blueberry.

**Funding:** The authors received no specific funding for this work.

**Competing interests:** The authors have declared that no competing interests exist.

## Introduction

Dormancy is an important adaption strategy for temperate fruit crops to survive harsh winter conditions. Buds require a certain amount of winter chill to break dormancy in spring. Since 1975, however, winter chill reductions have been recorded in temperate regions [1]. Under current global warming scenarios, dramatic winter chill reductions are projected to occur worldwide [2–6]. Inadequate winter chill is a major threat to the sustainability of temperate fruit crop production [7]. Artificial budbreak induction could be an important strategy for temperate fruit production in both subtropical and temperate climates.

Blueberry (*Vaccinium* spp.) is a temperate berry crop rich in antioxidants. Its production has rapidly expanded in recent years [8,9]. From 2009 to 2019, global blueberry production increased by 143%, mostly in countries with adequate winter chill, including the United States and Canada [9]. Blueberry production also expanded into subtropical and tropical regions [10,11], driven by high demand from new markets, improved cultivars, and innovative horticultural practices [12]. Southern highbush blueberry cultivars (*Vaccinium corymbosum* L. interspecific hybrids) with low chilling requirements are grown predominantly in subtropical regions [12], but their budbreak is still highly affected by inadequate winter chill [13,14]. For example, in the southeastern region of the United States (i.e., Florida), although low-chill Southern highbush blueberry cultivars can reach a high percentage of budbreak under natural conditions, the progress of natural budbreak is slow and prolonged. Because price premiums are closely associated with fruit earliness, many blueberry growers in this production region use hydrogen cyanamide (HC) as a bud dormancy-breaking agent to synchronize budbreak and advance fruit ripening [15,16].

Although HC is an important management tool for subtropical blueberry production [14], its use is strictly regulated or even prohibited in some countries because of its adverse effects on human health and the environment [17,18]. An ideal alternative to HC should not only be highly effective, but also safe and inexpensive. Chemical defoliation is a cultural practice that helps break bud dormancy for temperate fruit crops grown in subtropical regions [19,20]. Some fertilizers, such as zinc sulfate (ZS), urea, and potassium thiosulfate (KTS), can also be used as defoliants to artificially promote budbreak. For example, in blackberry (*Rubus* subgenus *Rubus* L. Watson) grown in West Central Florida, budbreak was induced most rapidly by urea application, followed by KTS, and ZS, advancing by 17 to 66 days compared with the control [21]. However, the efficacy of these defoliants in breaking bud dormancy has not been reported for blueberry.

Bud dormancy is regulated by the complex interaction of phytohormones [22]. Antagonistic roles of abscisic acid (ABA) and gibberellins (GA) are particularly well-documented [22,23]. In many temperate fruit crops, ABA concentrations increase at the onset of bud dormancy and decline during chill accumulation [24,25], whereas GA shows the opposite trend [26,27]. Jasmonic acid (JA) is also involved in dormancy release. For example, Juvany et al. [28] reported that JA accumulated during the transition from dormancy to budbreak in beech (*Fagus sylvatica* L.). Hao et al. [29] found that, in tea (*Camellia sinensis* L. O. Kuntze), the expression of JA synthesis genes was down-regulated during dormancy but up-regulated during budbreak. Indole-3-acetic acid (IAA) is reported to remain low during dormancy but increase during budbreak in Japanese pear (*Pyrus pyrifolia* Nakai) [30] and Japanese apricot (*Prunus mume*) [25]. Similar phytohormone dynamics also occur in HC-induced budbreak, such as increased GA and IAA accumulation and ABA degradation [31–34]. In blueberry, however, no study has quantified phytohormones in buds or investigated their responses to dormancy-breaking agents.

The objective of this study was to compare the effects of HC and three defoliants (KTS, urea, and ZS) applied at the beginning of chill accumulation on defoliation, flower budbreak, fruit ripening, and yield of 'Emerald' blueberry grown under subtropical climate conditions. Phytohormone profiling was performed to understand the potential roles of phytohormones in flower budbreak of blueberry, and to provide insight into the modes of action of HC and the defoliants.

## Materials and methods

### Experiment site and plant material

Two field experiments were conducted at a commercial blueberry farm located in Wimauma, Florida, United States (lat. 27˚42'N, long. 82˚17' W; elevation 31 m) during the 2018–2019 and 2019–2020 seasons. Plant material was 7-year-old 'Emerald' Southern highbush blueberry. The plants were grown in raised beds filled with pine bark, and they were spaced at 0.6 m within a row and 3.0 m between rows (5382 plants/ha). The number of accumulated chilling hours below 7.2˚C and chill portions recorded at the experiment site was obtained from the Florida Automated Weather Network (http://agroclimate.org/tools/chill-hours-calculator/).

### Defoliant treatment and experiment design

All treatments were applied between 9:00 and 11:00 AM with an airblast sprayer. A nonionic surfactant (Agri-Dex, Helena Chemical Co., Collierville, Tennessee, USA) was added at 0.5% (v/v) to all treatments including the water control in all the experiments. HC solutions were prepared using Dormex (50% hydrogen cyanamide; AlzChem, Trostberg, Germany).

In Expt. 1, treatments included water control, HC at 1.0% (active ingredient at 9.35 L ha$^{-1}$ or 1.74 mL/plant) and three defoliants, including potassium thiosulfate (KTS), urea, and zinc sulfate (ZS) at 6.0% (28 kg ha$^{-1}$ or 5.20 g/plant). The spray volume was 935 L ha$^{-1}$ for HC and 468 L ha$^{-1}$ for water and defoliants. All spray treatments were performed on 26 December 2018. The treatments were arranged in a randomized complete block design. All treatments had four replicated plots (one plot per block), except that the ZS treatment had three replicated plots. Each experimental unit (plot) consisted of 20 plants.

In Expt. 2, treatments included water control and 1.0% HC (active ingredient at 9.35 L ha$^{-1}$ or 1.74 mL/plant) sprayed at 935 L ha$^{-1}$ on 17 December 2019. The treatments were arranged in a completely randomized design, in which the water control and HC treatments had four and three replicated plots, respectively. Each experimental unit (plot) consisted of 20 plants.

In Expt. 3, treatments included water control with the spray volume at 1870 L ha$^{-1}$ and six ZS treatments. The six ZS treatments were derived from a factorial combination of two concentrations (5% or 10%) and three spray volumes (935, 1403, or 1870 L ha$^{-1}$), resulting in five ZS application rates (47, 70, 94, 141, and 187 kg ha$^{-1}$ or 8.7, 12.0, 17.5, 26.2, and 34.7 g/plant, respectively). All seven treatments were arranged in a randomized complete block design with five replicated plots (one plot per block). Each experimental unit (plot) consisted of 20 plants.

### Defoliation and flower budbreak

In Expt. 1 and 3, the number of leaves and flower buds were counted from the eight representative shoots selected from the six plants located in the middle of each plot before treatment. For each shoot, a section containing 30 to 40 nodes, was labeled to monitor defoliation and budbreak on a mostly weekly basis. The percentage of defoliation was calculated by dividing the number of nodes without leaves by the total number of nodes and multiplying by 100. Flower buds were considered sprouted when bud scales are separated at the tip—Stage 3 [35].

The percentage of flower budbreak was calculated by dividing the number of sprouted flower buds by the total number of flower buds and multiplying by 100.

## Yield and berry fresh weight

In Expt. 1 and 2, all ripe berries were harvested weekly from the six plants in the middle of each plot. At each harvest, we randomly selected fifty berries, weight them to determine the average berry fresh weight.

## Phytohormone extraction and profiling

Buds were sampled from the water control, HC, and ZS treatments at 8 days before treatment (DBT), 1, 9, and 20 days after treatment (DAT). Buds sampled at 8 DBT were pooled, and only the composite sample was used for phytohormone analysis. Buds sampled at 1, 9, and 20 DAT were stored and analyzed individually for each treatment. Collected samples were frozen in liquid nitrogen and stored at -80˚C for about six months. The frozen tissues were ground in liquid nitrogen to a fine powder using the mortar and pestle, and quickly weighed ($\approx$100 mg) into an 1.5 mL eppendorf tube. Phytohormone extraction and quantification were performed following the methods described by Almeida-Trapp et al. [36]. In brief, cold methanol:water (70:30, v/v) was immediately added to the samples. The samples were vortexed and sonicated, then extracted at 4˚C for 30 minutes, and centrifuged at 16000 x g at 4˚C for 5 min. The supernatant was removed and dried with nitrogen. Each sample was redissolved in 100% methanol plus labeled standards and the supernatant injected into a Waters Acquity I class UPLC connected to a Waters Xevo TQ-XS mass spectrometer (Waters Co., Milford, USA). Separation was carried out using the setting described in Lin and Agehara [38]. Cone and collision energy were optimized for each hormone individually; IAA and $d_5$IAA were analyzed in the positive ion mode while negative mode was used for JA, $d_5$JA, ABA, $d_6$ABA, $GA_4$, $^2H_2GA_4$, $GA_3$, and $^2H_2GA_3$. The selected reaction monitoring (SRM) analysis conditions were optimized for each phytohormone and internal standard. TargetLynx XS software (version 4.2; Waters Co., Milford, USA) was used to quantify peak area and the amount of constitutive hormone was based on comparison to labeled hormones.

## Statistical analysis

To identify the best-fit model for temporal responses of defoliation and flower budbreak to hydrogen cyanamide (HC) and zinc sulfate (ZS) in Expt. 1, the following four models were fit to each data set using SigmaPlot, and the best model was selected based on the smallest corrected Akaike information criterion (AICc).

Linear:

$$y = a + bx \qquad \text{Eq [1]}$$

Quadratic:

$$y = a + bx + cx^2 \qquad \text{Eq [2]}$$

Exponential plateau:

$$y = a + b[1 - \exp(-kx)] \qquad \text{Eq [3]}$$

Sigmoidal:

$$y = a/[1 + \exp(-(x - x_0)/b)] \qquad \text{Eq [4]}$$

In Eqs [1] and [2], $a$ is the $y$ intercept, $b$ is the linear coefficient, and $c$ is the quadratic coefficient. In Eq [3], $a$ is the $y$ intercept, $b$ is the maximum increase in $y$, and $k$ is the rate constant. In Eq [4], $a$ is the upper asymptote, $x_0$ is the $x$ value when $y$ reaches the midpoint between the baseline and maximum, and $b$ is the rate constant. In Eq [3], the sum of $a$ and $b$ is the upper asymptote, which represents the estimated maximum defoliation or budbreak. In Eq [4], $a$ represents the estimated maximum defoliation or budbreak, and $x_0$ represents the timing when HC or ZS induces a response (defoliation or budbreak) halfway between the baseline and maximum. When the same model was selected for more than two treatments, the model coefficients were compared between treatments. The model coefficients were deemed significantly different between two treatments ($P < 0.05$) if the 95% confidence intervals did not overlap.

All data were analyzed by the generalized linear mixed model procedure (PROC GLIMMIX) in SAS statistical software (SAS 9.4; SAS Institute Inc., Cary, NC, United States). Continuous data (yield and berry fresh weight), count data (berry number), and repeated measures (defoliation, budbreak, and phytohormone) were analyzed as described in Lin and Agehara [37]. Five different covariance structures, including compound symmetry (TYPE = CS), spatial Gaussian [(TYPE = SP(GAU)], spatial power [(TYPE = SP(POW)], spatial spherical [(TYPE = SP(SPH)], and unstructured covariance (TYPE = UN), were compared using maximum likelihood estimation with Laplace approximation (METHOD = LAPLACE) and default bias-corrected sandwich estimators (EMPIRICAL = MBN). The appropriate covariance structure was selected based on the smallest AICc. Then, model parameters were estimated using the restricted subject pseudo-likelihood method (METHOD = RSPL), and degrees of freedom for the fixed effects were adjusted using Kenward-Roger degrees of freedom approximation (DDFM = KR2).

For percentage data and count data, data were rescaled to the original scale by using the inverse link option (ILINK) in the LSMEANS statement. For phytohormone data, means and standard errors were back-transformed to the original scale using the Delta method. Least square means comparisons were performed using the Tukey-Kramer test. Unless otherwise noted, $P$ values $< 0.05$ were considered statistically significant. Rescaled or back-transformed data are reported in this study.

## Results

### Patterns of defoliation and budbreak (Expt. 1)

The progression of defoliation was fit to two different models in Expt. 1 (Fig 1). In the water control and urea treatments, defoliation followed linear models ($r^2 = 0.45$–0.53) with slopes of 0.63 to 0.65. No significant difference was found in both slope and intercept values between the two treatments. In the HC, KTS and ZS treatments, defoliation was described by an exponential plateau function, with a rapid increase from 15 DBT to 9 DAT followed by a gradual increase. The HC treatment had a higher rate constant value than the ZS and KTS treatments (0.03–0.07 vs. 0.14). According to the upper asymptote values, the maximum defoliation was estimated to be 98.6%, 75.6%, and 69.6% for the HC, ZS and KTS treatments, respectively.

The pre-treatment chill accumulation was 50 hours (8 chill portions), and the total chill accumulation, recorded from 1 November 2018 to 30 March 2019, was 145 hours (18 chill portions) in the first growing season (S1 Fig). The progression of budbreak was fit to two different models in Expt. 1 (Fig 1). In the water control and all defoliant treatments, budbreak followed a sigmoidal model. The exponential (rapid growth) phase in the ZS treatment occurred from

| Treatment | Defoliation | | | | | Budbreak | | | | | |
|---|---|---|---|---|---|---|---|---|---|---|---|
| | Best-fit model | $r^2$ | $a$ | $b$ | $k$ | Best-fit model | $r^2$ | $a$ | $b$ | $k$ | $x_0$ |
| Control | Linear | 0.45 | 32.03 | 0.65 | na | Sigmoidal | 0.91 | 83.70 | 9.41 | na | 37.69 b |
| HC | Exponential plateau | 0.99 | 89.02 a | 9.53 | 0.14 a | Exponential plateau | 0.83 | −84.27 | 163.26 | 0.12 | na |
| KTS | Exponential plateau | 0.66 | 34.13 c | 35.42 | 0.03 c | Sigmoidal | 0.95 | 76.03 | 7.45 | na | 37.60 b |
| Urea | Linear | 0.53 | 31.72 | 0.63 | na | Sigmoidal | 0.86 | 80.80 | 10.19 | na | 38.11 b |
| ZS | Exponential plateau | 0.97 | 55.47 b | 20.17 | 0.07 b | Sigmoidal | 0.93 | 71.74 | 9.77 | na | 26.95 a |

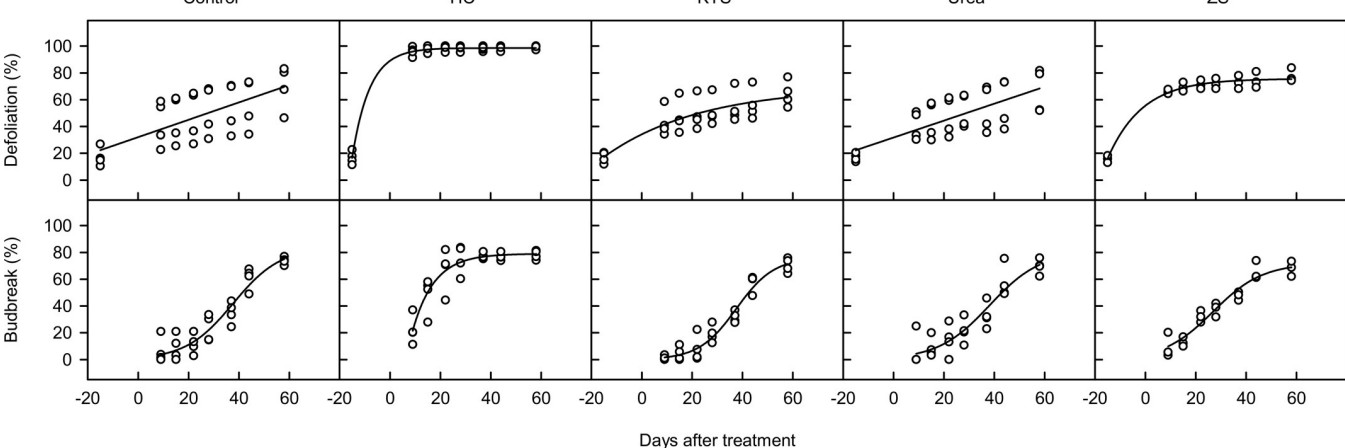

**Fig 1. The best-fit model for temporal responses of defoliation and flower budbreak to hydrogen cyanamide (HC) and defoliants in 'Emerald' blueberry grown under inadequate chilling conditions in the 2018–2019 season (Expt. 1).** Four models were fit to each data set: $y = a + bx$ (linear), $y = a + bx + cx^2$ (quadratic), $y = a + b[1 - exp(-kx)]$ (exponential plateau), and $y = a/[1 + exp(-(x-x_0)/b)]$ (sigmoidal). The best model was selected based on the smallest corrected Akaike information criterion. When the same model was selected for more than two treatments, the model coefficients were compared between treatments using 95% confidence intervals. The model coefficients in a column followed by the same letter or no letter are not significantly different ($P < 0.05$). KTS = potassium thiosulfate; ZS = zinc sulfate.

15 to 44 DAT, but that in the water control, KTS and urea treatments began later, occurring from 22 to 44 DAT. The rate constant and upper asymptote values showed no significant difference among the treatments. According to the $x_0$ values, the ZS treatment reached 50% of the maximum budbreak 10.74 days earlier than the water control (37.69 vs. 26.95 DAT). By contrast, other two defoliants did not show different $x_0$ values compared to the water control. In the HC treatment, budbreak was described by an exponential plateau function, with a rapid increase from 9 to 22 DAT followed by a gradual increase. According to the upper asymptote value, the maximum budbreak was estimated to be 79.0%.

## Defoliant effects on berry number, berry fresh weight, and yield (Expt. 1)

In the water control, harvest occurred between 105 and 131 DAT with a peak (1.50 t ha$^{-1}$) at 118 DAT (Fig 2). All defoliant treatments showed similar yield distribution as the water control, except that the HC treatment produced ripe berries at 100 DAT (0.63 t ha$^{-1}$), advancing harvest by 5 days. Compared with the water control, the HC treatment increased weekly yield by 125% to 311% throughout the harvest period. The yield increase by HC was statistically significant at 118 DAT (1.50 vs. 3.37 t ha$^{-1}$). The defoliant treatments had no significant effect on weekly yield, although the ZS treatment had 23% to 130% higher weekly yield than the water control throughout the harvest period.

Berry number was increased only by the HC treatment by 194% compared with the water control (433 vs. 1271 berries/plant) (Table 1). Among defoliant treatments, the ZS treatment had the highest berry number. Although the ZS treatment had 49% higher berry number than

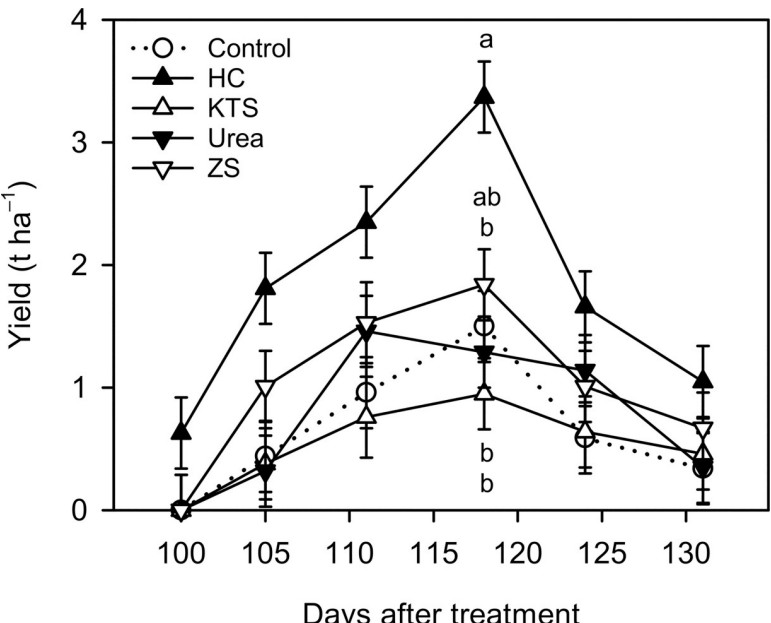

**Fig 2. Weekly yield of 'Emerald' blueberry grown under inadequate chilling conditions as affected by hydrogen cyanamide (HC) and defoliants in the 2018–2019 season (Expt. 1).** Treatments are as described in Fig 1. Means (n = 3–4) with the same or no letter within each measurement day are not significantly different (Tukey–Kramer test, $P < 0.05$). KTS = potassium thiosulfate; ZS = zinc sulfate.

the water control (433 vs. 644 berries/plant), the difference was not statistically significant. Berry fresh weight was not significantly affected by the treatments. Yield showed similar treatment responses compared to berry number. The HC treatment increased yield by 171% compared with the water control (3.70 vs. 10.03 t ha$^{-1}$). The ZS treatment had 41% higher yield than the water control (3.70 vs. 5.23 t ha$^{-1}$), but the difference was not statistically significant. Yield had a strong positive correlation with berry number ($r^2 = 0.98$), but it had no significant correlation with berry fresh weight (S2 Fig).

## Defoliant effects on bud development and phytohormone dynamics (Expt. 1)

From 1 to 9 DAT, buds appeared to be at the same development stage in all treatments with visible swelling and closed bud scales (Fig 3). In the water control, buds continued to swell but

**Table 1. Berry number, fresh weight (FW), and yield of 'Emerald' blueberry grown under inadequate chilling conditions as affected by hydrogen cyanamide (HC) and defoliants in the 2018–2019 season (Expt. 1).**

| Treatment[a] | Berry no. number | | Berry FW | Yield | |
|---|---|---|---|---|---|
| | (no./plant) | | (g/berry) | (t ha$^{-1}$) | |
| Control | 433 | b[c] | 1.63 | 3.70 | b |
| HC | 1271 | a | 1.49 | 10.03 | a |
| KTS | 367 | b | 1.52 | 2.91 | b |
| Urea | 538 | b | 1.64 | 4.69 | ab |
| ZS | 644 | ab | 1.60 | 5.23 | ab |
| P value[b] | 0.000 | | 0.680 | 0.004 | |

KTS = potassium thiosulfate; ZS = zinc sulfate.

[a]Treatments are as described in Fig 1.

[b]P values indicate the significance of the treatment effect.

[c]Means (n = 3–4) in a column followed by the same letter or no letter are not significantly different (Tukey–Kramer test, $P < 0.05$).

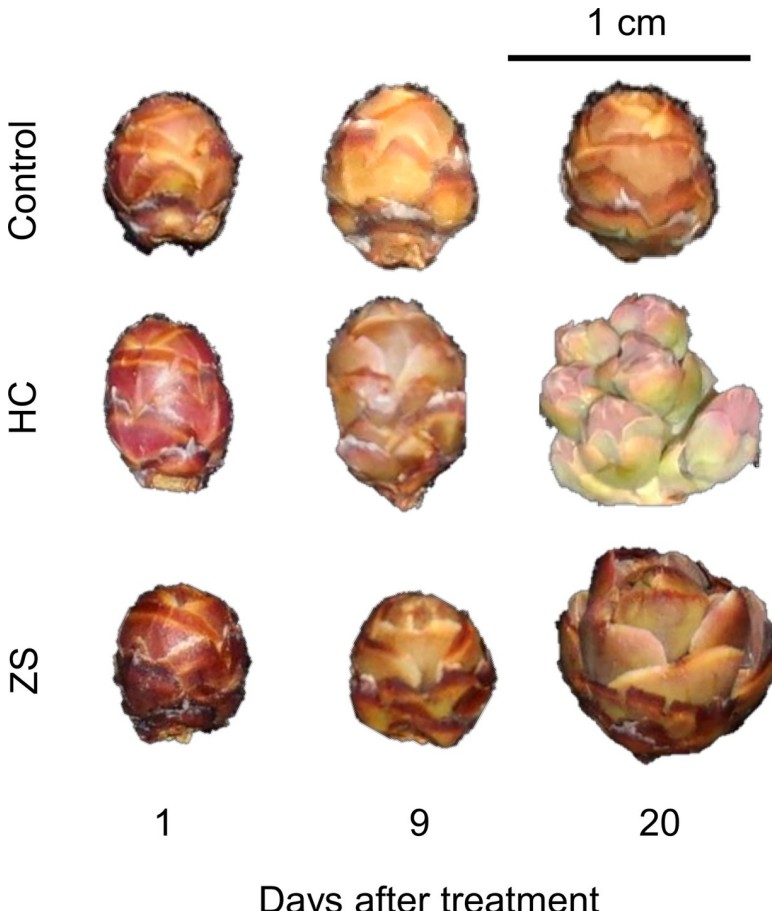

**Fig 3. Flower bud development of 'Emerald' blueberry grown under inadequate chilling conditions as affected by hydrogen cyanamide (HC) and zinc sulfate (ZS) in the 2018–2019 season (Expt. 1).** Treatments are as described in Fig 1.

remained at the same stage until 20 DAT. By contrast, bud development progressed in the HC and ZS treatments. At 20 DAT, HC-treated buds showed bud burst with individual flowers, whereas ZS-treated buds showed separation of bud scales but not bud burst. According to the classification method developed by Spiers (1978), the water control, HC-treated, and ZS-treated buds at 20 DAT were classified as Stage 2, 4, and 3, respectively.

The concentration of ABA at 8 DBT was 460 ng g$^{-1}$ (Fig 4A). In the water control, the ABA concentration remained constant until 20 DAT. In the HC treatment, the ABA concentration declined rapidly to 39% of the pre-treatment level at 1 DAT (166 ng g$^{-1}$), remained low until 9 DAT, and recovered to the water control level at 20 DAT. Compared with the water control, this transient HC-induced reduction in ABA was 59% to 63% (453–575 vs. 166–238 ng g$^{-1}$). By contrast, the ZS treatment had no significant effect on the ABA concentration throughout the measurement period.

The concentration of GA$_4$ at 8 DBT was 1.46 ng g$^{-1}$ (Fig 4B). In the water control and HC treatments, the GA$_4$ concentration increased transiently up to 3.34 ng g$^{-1}$ at 9 DAT. However, GA$_4$ had large standard errors, and no significant difference was detected among the treatments throughout the measurement period. In addition, the abundance of GA$_3$ was below our level of detection.

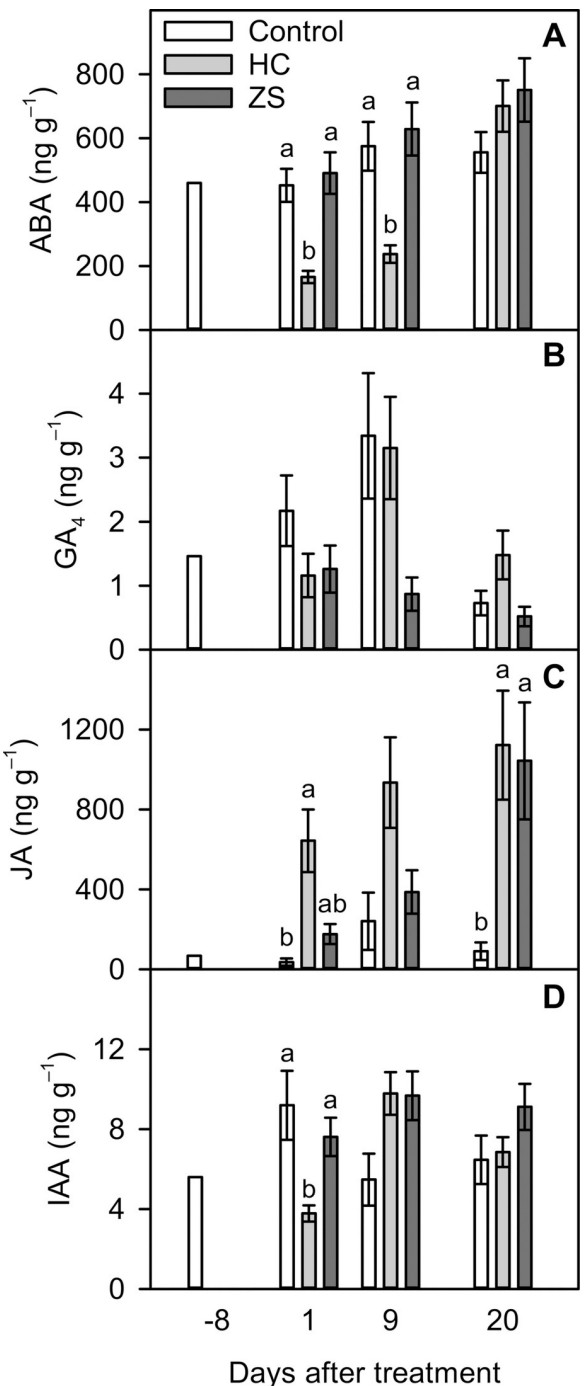

**Fig 4. Phytohormone dynamics in flower buds of 'Emerald' blueberry grown under inadequate chilling conditions as affected by hydrogen cyanamide (HC) and zinc sulfate (ZS) in the 2018–2019 season (Expt. 1).** **(A)** Abscisic acid (ABA). **(B)** Gibberellin 4 ($GA_4$). **(C)** Jasmonic acid (JA). **(D)** Indole-3-acetic acid (IAA). Treatments are as described in Fig 1. Pre-treatment samples (8 days before treatment) were pooled as a composite control sample. Means (n = 3–4) with the same or no letter within each measurement day are not significantly different (Tukey–Kramer test, $P < 0.05$).

The concentration of JA was at 8 DBT 69 ng g$^{-1}$ (Fig 4C). In the water control, the JA concentration rapidly increased by 141% at 9 DAT (241 ng g$^{-1}$) and declined to the pre-treatment level at 20 DAT. In the HC treatment, the JA concentration showed a sharp increase at 1 DAT (644 ng g$^{-1}$) followed by a gradual increase, resulting in a 15-fold increase from the pre-treatment level at 20 DAT (1122 ng g$^{-1}$). In the ZS treatment, the JA concentration showed a gradual increase until 9 DAT (388 ng g$^{-1}$) followed by a sharp increase, resulting in a 14-fold increase from the pre-treatment level at 20 DAT (1044 ng g$^{-1}$). Compared with the water control, the HC treatment increased the JA concentration 16- and 11-fold at 1 and 20 DAT, respectively, whereas the ZS treatment increased the JA concentration 10-fold at 20 DAT.

The concentration of IAA at 8 DBT was 5.61 ng g$^{-1}$ (Fig 4D). In the water control, the IAA concentration remained constant until 20 DAT (5.48–9.20 ng g$^{-1}$). In the HC treatment, the IAA concentration showed a transient reduction at 1 DAT (3.78 ng g$^{-1}$), but it recovered to the pre-treatment level thereafter. Compared with the water control, the HC treatment significantly decreased the IAA concentration by 59% at 1 DAT. The ZS treatment had no significant effect on the IAA concentration throughout the measurement period.

## HC effects on berry number, fresh weight and yield (Expt. 2)

In the water control, harvest occurred between 112 and 134 DAT with a peak (2.02 t ha$^{-1}$) at 119 DAT (Fig 5). The HC treatment produced ripe berries at 105 DAT (3.41 t ha$^{-1}$), advancing harvest by 7 days. From 112 to 127 DAT, the HC treatment increased weekly yield by 89% to 113% compared with the water control. The yield increase by HC was statistically significant at 105 DAT (0.00 vs. 3.41 t ha$^{-1}$) and 119 DAT (2.02 vs. 3.81 t ha$^{-1}$).

Compared with the water control, the HC treatment increased berry number by 246% (295 vs. 1021/plant), decreased the berry fresh weight by 17% (1.97 vs. 1.63 g/berry), and increased yield by 183% (3.13 vs. 8.86 t ha$^{-1}$) (Table 2).

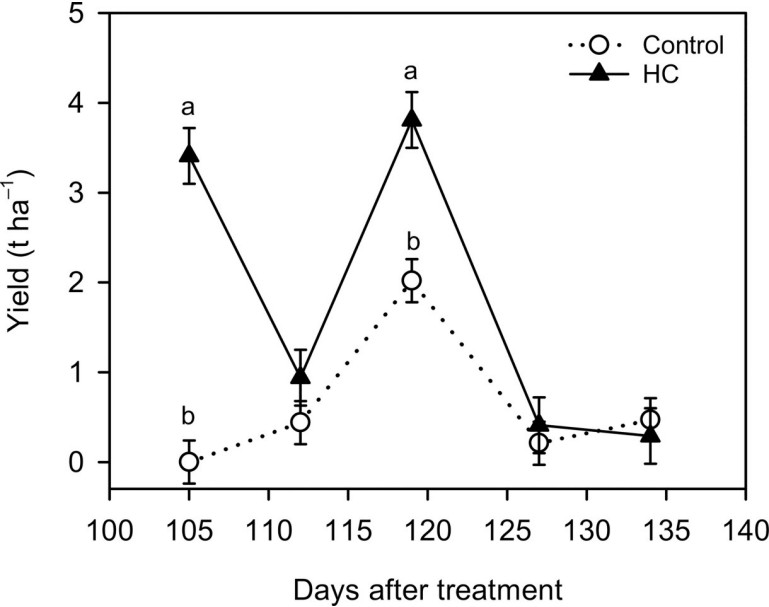

**Fig 5. Weekly yield of 'Emerald' blueberry grown under inadequate chilling conditions as affected by hydrogen cyanamide (HC) in the 2019–2020 season (Expt. 2).** Treatments included water control and 1.0% HC (9.35 L ha$^{-1}$ or 1.74 mL/plant) sprayed at 935 L ha$^{-1}$ on 17 December 2019. A non-ionic surfactant (Agri-Dex) was added at 0.5% (v/v) to both water control and HC treatments. Means (n = 3–4) with the same or no letter within each measurement day are not significantly different (Tukey–Kramer test, $P < 0.05$).

**Table 2. Berry number, fresh weight (FW), and yield of 'Emerald' blueberry grown under inadequate chilling conditions as affected by hydrogen cyanamide (HC) in the 2019–2020 season (Expt. 2).**

| Treatment[a] | Berry number (no./plant) | | Berry FW (g/berry) | | Yield (t ha$^{-1}$) | |
|---|---|---|---|---|---|---|
| Control | 295 | a[c] | 1.97 | a | 3.13 | b |
| HC | 1021 | b | 1.63 | b | 8.86 | a |
| *P* value[b] | 0.002 | | 0.010 | | 0.001 | |

[a]Treatments are as described in Fig 5.

[b]*P* values indicate the significance of the treatment effect.

[c]Means (n = 3–4) in a column followed by the same letter or no letter are not significantly different (Tukey–Kramer test, *P* < 0.05).

## ZS concentration and spray volume effects on defoliation and budbreak (Expt. 3)

Defoliation barely initiated at 9 DBT, with the percentage of defoliation ranging from 1.4% to 3.1% at 9 DBT (Fig 6). In the water control, defoliation slowly increased from 8.1% at 10 DAT to 21.1% at 59 DAT. All ZS treatments showed similar defoliation responses as the water control. At 59 DAT, the final percentage of defoliation ranged from 13.4% to 20.4% in all ZS treatments.

The pre-treatment chill accumulation was 14 hours (6 chill portions), and the total chill accumulation, recorded from 1 November 2019 to 30 March 2020, was 111 hours (14 chill portions) in the second growing season (S1 Fig). Budbreak was not observed when the treatments were applied on 20 December 2019. In the water control, budbreak was limited from 10 to 33 DAT (1.2%–10.0%), but increased gradually from 19.8% at 47 DAT to 75.8% at 83 DAT (Fig 6). All ZS treatments showed similar budbreak responses from 10 to 83 DAT. At 83 DAT, the final percentage of budbreak ranged from 81.9% to 88.7% in all ZS treatments.

## Discussion

### Chill-dependent and -independent budbreak induction

Dormancy-breaking agents and defoliants can promote budbreak directly or indirectly. HC is a highly toxic compound to plants. HC can inhibit catalase activity [38] and impair mitochondrial function [39], which in turn increases reactive oxygen species (ROS) accumulation [40]. Some previous studies suggest that ROS production plays a direct role in HC-induced budbreak in temperate fruit crops [40–42]. For example, in grapevine (*V. vinifera.* × *V. labruscana*), HC rapidly increased $H_2O_2$ production in buds after 12 hours of treatment and increased budbreak from 0% to 53% compared with the water control at 8 DAT [43]. On the other hand, defoliation can indirectly promote budbreak by enhancing the receptibility of buds to winter chilling and fulfilling chilling requirements [44,45]. In this study, both defoliation and budbreak in the HC treatment were described by an exponential plateau function with a rapid increase from 0 to 22 DAT, during which chill accumulation increased only by 12 hours. This observation suggests that HC-induced budbreak is facilitated by rapid regulation of signaling molecules, such as ROS and phytohormones, rather than by increased receptibility of buds to winter chilling. Potential roles of phytohormones in HC-induced budbreak are discussed in Discussion below ("Potential roles of ABA and GA in HC-induced budbreak" and "Antagonistic interactions between JA and ABA and effects of prolonged JA accumulation on flower development").

Model fitting analysis revealed that only HC and ZS advanced both defoliation and budbreak compared with the water control. Therefore, the discussion below is primarily about

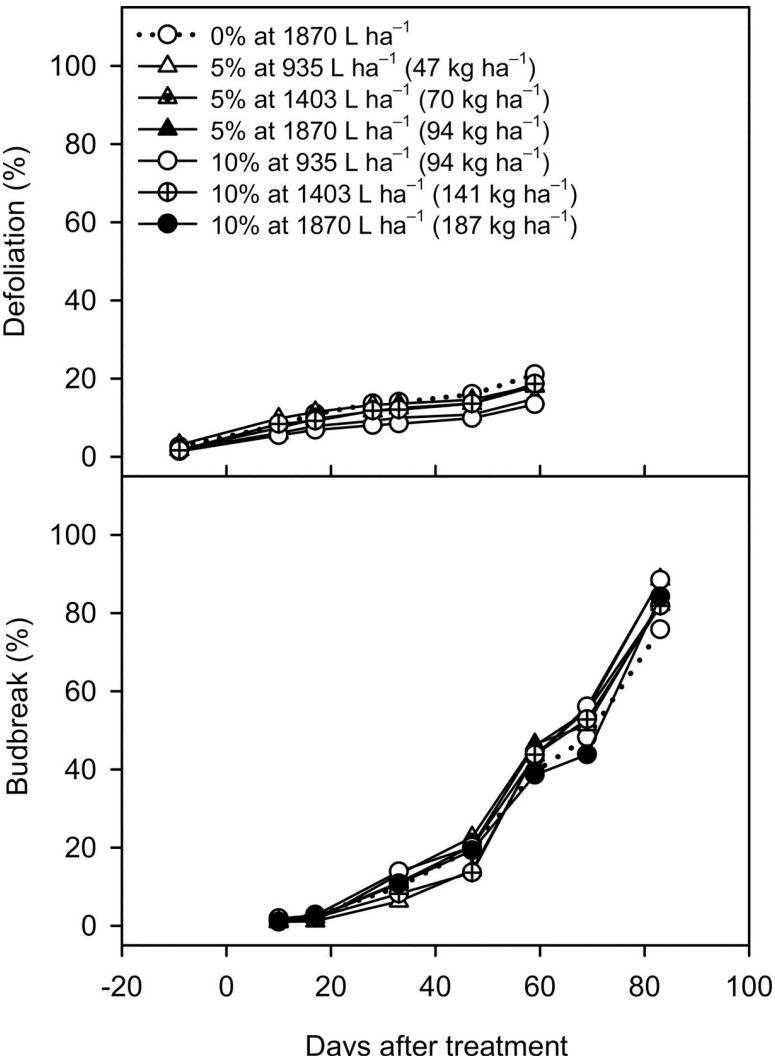

**Fig 6. Defoliation and flower budbreak of 'Emerald' blueberry grown under inadequate chilling conditions as affected by different application rates of zinc sulfate (ZS) in the 2019–2020 season (Expt. 3).** Treatments included water control at 1870 L ha$^{-1}$ and six ZS treatments in a factorial combination of two concentrations (5% or 10%) and three spray volumes (935, 1403, or 1870 L ha$^{-1}$ or or 8.7, 12.0, 17.5, 26.2, and 34.7 g/plant) on 20 December 2019. A non-ionic surfactant (Agri-Dex) was added at 0.5% (v/v) to all treatments including the water control. Means (n = 5) with the same or no letter within each measurement day are not significantly different (Tukey–Kramer test, $P < 0.05$).

these two treatments. Interestingly, in the first growing season, although both HC and ZS induced rapid defoliation in an exponential plateau function, they induced different budbreak responses: HC-induced budbreak was a rapid exponential plateau function, whereas ZS-induced budbreak was a gradual sigmoidal function. Compared with HC-induced break, the rapid phase of ZS-budbreak occurred over a prolonged period from 15 to 44 DAT, during which chill accumulation increased by 67 hours. Consequently, ZS-induced budbreak appeared to occurr in parallel with chill accumulation. These results suggest that HC and ZS have different modes of action in budbreak induction. As discussed above, HC-induced bud-break may be facilitated by the rapid regulation of signaling molecules. By contrast, the pattern of ZS-induced budbreak suggests that its mode of action involves increased receptibility of buds to winter chilling.

It is also interesting to note that, in this study, ZS accelerated defoliation and budbreak only in the first season. Pre-treatment chill accumulation was 36 hours greater in the first growing season than in the second growing season (50 vs. 14 hours), whereas post-treatment chill accumulation was similar in the two growing seasons (95 to 97 hours), suggesting that the efficacy of ZS is dependent on pre-treatment chill accumulation. Furthermore, the HC treatment had 13% higher yield in the first season than in the second season. Chill-dependent efficacy of defoliants is also reported in previous studies. Bi et al. [46] found that delaying ZS application from early November to late December increased the percentage of defoliation from 5% to 100% in almond (*Prunus dulcis* (Mill) D. A. Webb). In an experiment with controlled environments, Stringer et al. [47] found that the efficacy of HC in advancing bud development of 'Bladen' blueberry was enhanced when pretreatment chill accumulation was increased from 300 to 450 hours. In blackberry, we examined the budbreak induction effects of lime sulfur over two growing seasons under climate conditions similar to this study. The final percentage of budbreak was 76.8% in the first season with 194 pretreatment chilling hours, but it was only 42.6% in the second season with 101 pre-treatment chilling hours [37].

## High efficacy of HC in advancing fruit ripening and increasing yield compared with other defoliants

The effectiveness of HC in advancing fruit ripening aligns with previous studies. In southern Uruguay, Arias et al. [48] reported that 1.0% of HC increased early-season yield by 8.0% to 13.2% in 'O'Neal' blueberry. In northern Florida, Williamson et al. [15] reported that 1.0% to 2.0% of HC advanced fruit ripening by 10 to 14 days in 'Misty' blueberry. In fact, the application of HC is a common commercial practice to advance budbreak and fruit ripening in central Florida [16]. Our results further confirm that HC is highly effective in advancing fruit ripening in blueberry grown under inadequate chilling conditions.

The average commercial blueberry yield in Florida from 2018 to 2019 was 4.87 t ha$^{-1}$, which was 69% of the national average yield (7.09 t ha$^{-1}$), according to the USDA's National Agricultural Statistics Service [49]. In this study, yields of HC-treated plants (8.86 to 10.03 t ha$^{-1}$) were 25% to 41% higher than the average yield in the United States and 82% to 106% higher than the average yield in Florida, demonstrating the high effectiveness of HC as a dormancy-breaking agent and its importance for subtropical blueberry production.

However, a negative side effect of HC was also documented in this study: HC increased berry number by 246% but reduced the average berry fresh weight by 17%, compared with the water control. A negative relationship between fruit number and weight is often found in small fruit crops [50–52]. Therefore, the negative impact of HC on the average berry fresh weight can be explained by source-sink relationships.

In contrast to HC, all defoliants tested in this study had no significant effect on yield. Although ZS promoted budbreak and defoliation to some extent in the first growing season, it showed no significant effect in the second growing season, regardless of the application rate. These results indicate limited and inconsistent efficacy of the defoliants in budbreak induction in blueberry. In particular, the fact that defoliation was unaffected by ZS even at an extremely high application rate of 187 kg ha$^{-1}$ (10% at 1870 L ha$^{-1}$) indicates high hardiness of the tested cultivar 'Emerald'.

## Potential roles of ABA and GA in HC-induced budbreak

Phytohormone dynamics mediate the control of bud dormancy and budbreak in temperate fruit crops [22]. In blueberry, this study, for the first time, performed phytohormone profiling

in flower buds during the transition from dormancy to budbreak. Our results, therefore, provide insight into the hormonal regulation of budbreak in blueberry.

Antagonistic roles of ABA and GA in the regulation of dormancy and budbreak are well-documented [41,53,54]. Using transgenic plants, several studies suggest that ABA and GA are involved in the metabolic and gene regulation of each other [22,55]. In general, the ABA concentrations in buds declines with the depth of dormancy [22,24,25,55,56], whereas GA shows an exact opposite trend [26,27]. This inverse correlation between ABA and GA is reported in many *Rosaceae* fruit crops [26,27,30,57], and it can also be induced by HC in grape (*Vitis vinifera* L.) and sweet cherry (*Prunus avium* L.) [31,32,34]. In this study, although the ABA concentration in HC-treated buds declined rapidly before the onset of budbreak, $GA_4$ showed no significant changes and $GA_3$ was undetectable. The lack of the ABA and GA antagonism suggests two possibilities. First, budbreak in blueberry may be regulated by ABA and other phytohormones independently of GA. Second, GA bioactive forms other than $GA_3$ and $GA_4$, such as $GA_1$ and $GA_7$, could be involved in the regulation of budbreak in blueberry. Accumulation of $GA_1$ and $GA_7$ has been reported to occur during dormancy release in buds of Japanese apricot (*Prunus mume*) [25] and peach (*Prunus persica* L.) [57].

Because ABA promotes bud dormancy and inhibits budbreak [58], high efficacy of HC in budbreak induction can be explained by HC-induced ABA reductions. Similar HC-induced ABA reductions have been reported in sweet cherry [32] and grapevine [31,34]. Some molecular studies on grapevine suggest that the inhibitory effect of HC on ABA accumulation involves both down-regulation of ABA biosynthesis genes and up-regulation of ABA degradation genes [31,34,59,60]. In blueberry, it is likely that HC rapidly reduces the ABA concentration in flower buds by mediating ABA metabolism. Our results also suggest that the effects of HC on ABA metabolism are transient and reversible.

## Antagonistic interactions between JA and ABA and effects of prolonged JA accumulation on flower development

In addition to ABA and GA, JA is involved in the regulation of budbreak in some perennial plants. Juvany et al. [28] reported that JA production increased during budbreak in beech. In a transcriptome study, Hao et al. [29] found that the expressions of JA synthesis genes were down-regulated during dormancy but up-regulated during budbreak in tea plants. In sweet cherry, Ionescu et al. [33] also reported up-regulation of JA synthesis genes during HC-induced budbreak. In this study, JA production in HC- and ZS-treated buds increased steadily in parallel with budbreak progression, suggesting that JA plays a promotive role in budbreak in blueberry.

Dormancy has common regulatory pathways in buds and seeds [31,61–63]. In apple, endogenous JA increased during stratification and exogenous JA promoted germination of embryos [64]. Some studies proposed that JA promotes seed dormancy release by repressing the expression of ABA biosynthesis genes in wheat [65,66]. Therefore, a transient decline in ABA in HC-treated buds could be mediated by JA accumulation.

Interestingly, the antagonistic relationship between JA and ABA occurred only in HC-treated buds, although both HC and ZS dramatically increased JA production at 20 DAT. Unlike HC-treated buds, ZS-treated buds did not show significant increases in JA production at 1 and 9 DAT. In ZS-treated buds, therefore, it is possible that the antagonistic interaction between JA and ABA could have happened between 9 and 20 DAT because of the delayed response in JA accumulation. In addition to JA accumulation, budbreak occurred relatively slowly in ZS-treated buds than in HC-treated buds, further indicating the importance of JA in budbreak induction.

It is also interesting to note that, in HC-treated buds, JA production remained high even when individual flowers became distinguishable in the flower cluster. In addition to budbreak, JA is known to promote floral organ development and flower opening [67,68]. Ishiguro et al. [69] found that an Arabidopsis mutant characterized by a delay in flower opening, *defective in anther dehiscence* (*dad1*), had a 78% lower concentration of jasmonates (the sum of JA and methyl JA) during flowering than the wildtype. They also reported that exogenous JA could restore normal flower opening in this mutant. In this study, therefore, it is possible that prolonged increases in JA production have promotive effects not only on budbreak but also on flower development, which in turn, could contribute to improved fruit earliness.

## IAA may not play an important role in floral development

As a growth-promoting phytohormone, IAA concentrations generally remain low during dormancy but increase rapidly after budbreak [25,30]. Liang et al. [34] reported that IAA production also increased during HC-induced budbreak in grapevine buds. In this study, however, IAA production did not increase in both HC- and ZS-treated buds after the onset of budbreak. Similarly to auxins (e.g. IAA), cytokinins are a class of phytohormones that promote cell division in plant shoots and roots [70]. In many perennial fruit crops, cytokinin production increases during natural [71] and HC-induced budbreak [33,34,71,72]. In blueberry, therefore, initial flower development may be facilitated by cytokinins, rather than auxins.

Interestingly, the IAA concentration in HC-treated buds showed a rapid decline, although it recovered to the pre-treatment level before budbreak. Ionescu et al. [33] reported a similar transient decline in IAA, as well as the down-regulation of an IAA biosynthesis enzyme in HC-treated sweet cherry. These contrasting IAA responses suggest that the effect of HC on IAA accumulation is dependent on plant species.

## Conclusion

Our results collectively suggest that defoliants are not effective alternatives to HC, and that HC- and ZS-induced budbreak have different modes of action as illustrated in Fig 7. In blueberry, we propose that HC treatment increases JA production, which in turn, decreases ABA concentration in buds by changing ABA homeostasis (i.e. inhibiting ABA production or promoting ABA catabolism). This antagonistic interaction between JA and ABA appears to be the primary mode of action for HC. Although ZS also increases JA production, it occurs relatively slower compared to HC-induced JA production. Therefore, we propose that ZS does not stimulate the antagonism between JA and ABA as effectively as HC, as indicated by the dashed line in Fig 7. Furthermore, we propose that ZS-induced budbreak is facilitated also by defoliation. This defoliation-mediated budbreak is a relatively slow process, as it involves increased receptibility to winter chilling, which in turn, helps fulfill chilling requirements earlier. These different modes of action between HC and ZS may explain the high efficacy of HC as a dormancy-breaking agent. On the other hand, low and inconsistent efficacy of ZS may be due to high hardiness of the tested blueberry cultivar. To better evaluate the potential of ZS as an alternative dormancy-breaking agent to HC, future research should test ZS for less hardy blueberry cultivars.

In temperate fruit crops, the harvest window, potential yield and fruit quality are determined by the timing, rate and uniformity of budbreak [73]. Under inadequate chill conditions, HC has been used extensively as a bud dormancy-breaking agent, but its hazardous properties lead to its strict regulations and ongoing exploration of safer alternatives to HC [41]. Furthermore, it is projected that global warming will cause dramatic winter chill reductions worldwide [2–6] and thus negatively affect the viability of temperate fruit crop production in the future

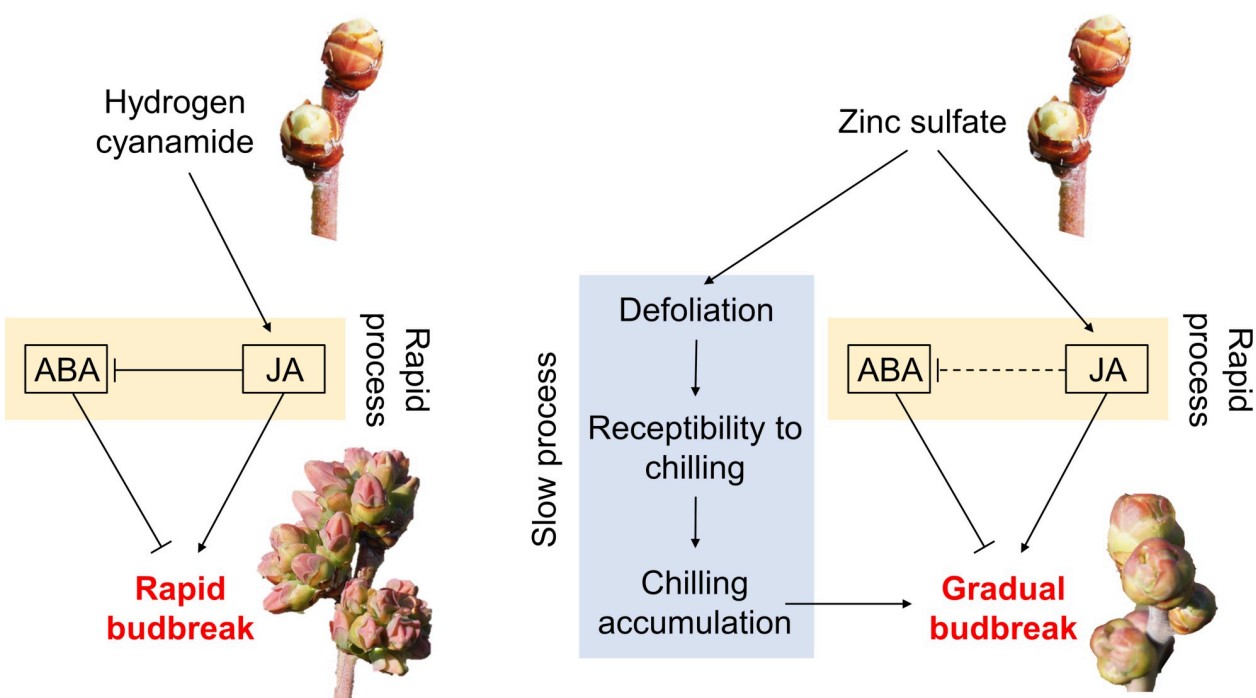

**Fig 7. Proposed modes of action for budbreak induction by hydrogen cyanamide and zinc sulfate in blueberry.** Solid lines indicate demonstrated or highly likely relations. Dashed lines indicate hypothetical interactions. ABA = abscisic acid; JA = jasmonic acid.

[7]. Therefore, the proposed modes of action of HC- and ZS-induced budbreak in this study not only provide the fundamental understanding of bud dormancy-breaking agents, but also help develop bud dormancy-breaking agents that improve the sustainability and resilience of temperate fruit production under changing environments.

## Supporting information

**S1 Fig. Chilling hours and chill portions at the experiment site in central Florida, USA in the 2018–2019 and 2019–2020 seasons.** The number of accumulated chilling hours below 7.2˚C recorded at the experiment site was obtained from the Florida Automated Weather Network (http://agroclimate.org/tools/chill-hours-calculator/).
(TIF)

**S2 Fig. Linear regression between berry number and yield or berry fresh weight (FW) and yield of 'Emerald' blueberry grown under inadequate chilling conditions in the 2018–2019 season (Expt. 1).** Treatments are as described in Fig 1. No line was included for the correlation between berry FW and yield because of non-significant correlation.
(TIF)

## Acknowledgments

We thank Astin Farms for allowing us to conduct experiments. We also thank Dr. Zhanao Deng, Dr. Seonghee Lee, Dr. Patricio Munoz, and Dr. Jeffrey Williamson for their scientific guidance and constructive discussion, and all members of Horticultural Crop Physiology Lab at the Gulf Coast Research and Education Center for their technical assistance.

## Author Contributions

**Conceptualization:** Shinsuke Agehara.

**Data curation:** Syuan-You Lin.

**Formal analysis:** Syuan-You Lin.

**Funding acquisition:** Shinsuke Agehara.

**Investigation:** Syuan-You Lin.

**Methodology:** Syuan-You Lin.

**Supervision:** Shinsuke Agehara.

**Writing – original draft:** Syuan-You Lin.

**Writing – review & editing:** Syuan-You Lin, Shinsuke Agehara.

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
