## [Decision Letter · Decision Letter 0]

29 Jul 2021

PONE-D-21-21254

Budbreak patterns and phytohormone dynamics reveal different modes of action between hydrogen cyanamide- and defoliant-induced flower budbreak in blueberry under inadequate chilling conditions

PLOS ONE

Dear Dr. Agehara,

Thank you for submitting your manuscript to PLOS ONE. After careful consideration, we feel that it has merit but does not fully meet PLOS ONE’s publication criteria as it currently stands. Therefore, we invite you to submit a revised version of the manuscript that addresses the points raised during the review process.

We look forward to receiving your revised manuscript.

Kind regards,

Chunxian Chen, Ph.D.

Academic Editor

PLOS ONE

Journal Requirements:

3. Please remove your supplementary figures from within your manuscript file, leaving only the individual TIFF/EPS image files, uploaded separately.  Please note that supplementary figures/tables should be uploaded as separate "supporting information" files."

Additional Editor Comments:

The paper provided useful results and understanding for use of HC and other defoliation agents on blueberry production with insufficient chill. Experiments were well designed and data well analyzed. However, Fig. 7 appeared to be overstretched and weak. If ZS also suggestively induces the ABA-JA rapid process as HC, it would difficult to explain how the effects are different between them. It would be more convincing and reasonable the process and responses to the two agents are rather very different from each other. In addition, there are reports on substantial phytotoxicity of HC to buds and new growth in other temperate crops, which might be a lead factor of yield loss. Were there data on phytotoxicity to blueberry from those experiments? If so, it should be included in the paper.

Reviewers' comments:

Reviewer's Responses to Questions

**Comments to the Author**

1. Is the manuscript technically sound, and do the data support the conclusions?

Reviewer #1: Partly

2. Has the statistical analysis been performed appropriately and rigorously? 

Reviewer #1: I Don't Know

3. Have the authors made all data underlying the findings in their manuscript fully available?

Reviewer #1: Yes

4. Is the manuscript presented in an intelligible fashion and written in standard English?

Reviewer #1: Yes

5. Review Comments to the Author

Reviewer #1: This manuscript brings forward valuable information for researchers and producers of temperate fruit crops such as blueberry. The information could have been presented in a more organized manner with greater clarity however. The description of the treatments was not very clear, which made it difficult to follow the results and discussion. It was not until I got to the conclusions that I was more certain of my understanding of the manuscript findings.

This study is particularly important to the fruit industry. Please make every effort to make the treatment descriptions and the findings more clear.

Line 44: Delete 'adaptation'.

Line 66-67: Is the budbreak mentioned in these citations a result in defoliation?

Line 103-104: Is this 1% of product or ai?

113-118: Description of the experimental treatments is unclear to say the least. What are the treatments - spray volume? concentration? rates?

Line 137: stored at -80 C for how long?

Line 204-208: This was previously stated in the m and m where it belongs.

Line 230: Treatment produced ripened berries.

Line 290: Treatment may be depicted by the wrong color in the graph. Please check.

Line 357: "...chill hour increased only by 12 hours." Not sure how this was determined.

Line 398-399: This is worded as though Florida is not a part of the United States. Please reword this.

Line 504: Please place comma after 'conditions'.

6. PLOS authors have the option to publish the peer review history of their article (what does this mean?). If published, this will include your full peer review and any attached files.

Reviewer #1: No

---

## [Author Response · Author response to Decision Letter 0]

17 Aug 2021

Additional Editor Comments:

1. The paper provided useful results and understanding for use of HC and other defoliation agents on blueberry production with insufficient chill. Experiments were well designed and data well analyzed. However, Fig. 7 appeared to be overstretched and weak. If ZS also suggestively induces the ABA-JA rapid process as HC, it would difficult to explain how the effects are different between them. It would be more convincing and reasonable the process and responses to the two agents are rather very different from each other.

ANS: Thank you for your suggestions! We think the modes of action of HC- and ZS-induced budbreak share some differences and similarities. We found that both HC and ZS increase JA production, but ZS induced JA accumulation more rapidly than HC. Therefore, we propose that ZS does not stimulate the antagonism between JA and ABA as effectively as HC (L488-490). To avoid presenting overstretched conclusion in Fig. 7, we used solid lines to demonstrate highly likely relations and dashed lines to indicate hypothetical interactions obtained from literature.

2. In addition, there are reports on substantial phytotoxicity of HC to buds and new growth in other temperate crops, which might be a lead factor of yield loss. Were there data on phytotoxicity to blueberry from those experiments? If so, it should be included in the paper.

ANS: We did these experiments in a commercial orchard and all HC treatments were performed following the instruction on Dormex’s label. Because we and growers did not observe any phytotoxicity during the experimental period, we think the lead factor of yield loss was due to the difference in chilling hours between the two seasons.

Reviewers' comments:

Reviewer's Responses to Questions

Reviewer #1: This manuscript brings forward valuable information for researchers and producers of temperate fruit crops such as blueberry. The information could have been presented in a more organized manner with greater clarity however. The description of the treatments was not very clear, which made it difficult to follow the results and discussion. It was not until I got to the conclusions that I was more certain of my understanding of the manuscript findings.

1. This study is particularly important to the fruit industry. Please make every effort to make the treatment descriptions and the findings more clear.

ANS: Thank you for your comments! We have made revisions in the manuscript to address all your suggestions.

2. Line 44: Delete 'adaptation'.

ANS: Revised as suggested (L44).

3. Line 66-67: Is the budbreak mentioned in these citations a result in defoliation?

ANS: In the almond study (Hegazi, 2012), defoliation data was not presented. Therefore, we decide to delete this citation. In the blackberry study (Lin and Agehara, 2021), urea, zinc sulfate and potassium thiosulfate (KTS) were sprayed as defoliants and defoliation percentage was recorded. The authors concluded that defoliation is one of many factors involving in budbreak induction (L65-67). 

4. Line 103-104: Is this 1% of product or ai?

ANS: It is 1% of ai (L101, L108).

5. L113-118: Description of the experimental treatments is unclear to say the least. What are the treatments - spray volume? concentration? rates?

ANS: We’ve revised the paragraph to clarify the treatments in Expt. 3 (L112-115).

6. Line 137: stored at -80 C for how long?

ANS: For about six months. We’ve added this information (L136).

7. Line 204-208: This was previously stated in the m and m where it belongs.

ANS: Deleted as suggested (L203).

8. Line 230: Treatment produced ripened berries.

ANS: Revised as suggested (L226).

9. Line 290: Treatment may be depicted by the wrong color in the graph. Please check.

ANS: We’ve checked it. Can you point out which colors are wrong please? (Fig. 4C)

10. Line 357: "...chill hour increased only by 12 hours." Not sure how this was determined.

ANS: The number of accumulated chilling hours below 7.2°C recorded at the experiment site was obtained from the Florida Automated Weather Network (http://agroclimate.org/tools/chill-hours-calculator/) (Supplementary Figure 1). The difference of chill hours between 22 and 0 DAT is determined by subtracting chill accumulation at 0 DAT from chill accumulation at 22 DAT (L352-353).

11. Line 398-399: This is worded as though Florida is not a part of the United States. Please reword this.

ANS: Revised as suggested (L394-396).

12. Line 504: Please place comma after 'conditions'.

ANS: Revised as suggested (L502).

---

## [Editor Report · Decision Letter 1]

19 Aug 2021

Budbreak patterns and phytohormone dynamics reveal different modes of action between hydrogen cyanamide- and defoliant-induced flower budbreak in blueberry under inadequate chilling conditions

PONE-D-21-21254R1

Dear Dr. Agehara,

We’re pleased to inform you that your manuscript has been judged scientifically suitable for publication and will be formally accepted for publication once it meets all outstanding technical requirements.

Kind regards,

Chunxian Chen, Ph.D.

Academic Editor

PLOS ONE
---

## [Editor Report · Acceptance letter]

23 Aug 2021

PONE-D-21-21254R1 

Budbreak patterns and phytohormone dynamics reveal different modes of action between hydrogen cyanamide- and defoliant-induced flower budbreak in blueberry under inadequate chilling conditions 

Dear Dr. Agehara:

I'm pleased to inform you that your manuscript has been deemed suitable for publication in PLOS ONE. Congratulations! Your manuscript is now with our production department. 

Kind regards, 

on behalf of

Dr. Chunxian Chen 

Academic Editor

PLOS ONE